# Deep Diversity: Extensive Variation in the Components of Complex Visual Systems across Animals

**DOI:** 10.3390/cells11243966

**Published:** 2022-12-08

**Authors:** Oliver Vöcking, Aide Macias-Muñoz, Stuart J. Jaeger, Todd H. Oakley

**Affiliations:** 1Department of Biology, University of Kentucky, Lexington, KY 40508, USA; 2Department of Ecology, Evolution, and Marine Biology, University of California, Santa Barbara, CA 93106, USA

**Keywords:** eye evolution, opsin, photoreceptor, phototransduction, visual cycle

## Abstract

Understanding the molecular underpinnings of the evolution of complex (multi-part) systems is a fundamental topic in biology. One unanswered question is to what the extent do similar or different genes and regulatory interactions underlie similar complex systems across species? Animal eyes and phototransduction (light detection) are outstanding systems to investigate this question because some of the genetics underlying these traits are well characterized in model organisms. However, comparative studies using non-model organisms are also necessary to understand the diversity and evolution of these traits. Here, we compare the characteristics of photoreceptor cells, opsins, and phototransduction cascades in diverse taxa, with a particular focus on cnidarians. In contrast to the common theme of deep homology, whereby similar traits develop mainly using homologous genes, comparisons of visual systems, especially in non-model organisms, are beginning to highlight a “deep diversity” of underlying components, illustrating how variation can underlie similar complex systems across taxa. Although using candidate genes from model organisms across diversity was a good starting point to understand the evolution of complex systems, unbiased genome-wide comparisons and subsequent functional validation will be necessary to uncover unique genes that comprise the complex systems of non-model groups to better understand biodiversity and its evolution.

## 1. Introduction

A leading question in convergent evolution is, how similar or different are the genes and regulatory mechanisms underlying complex traits between species? Investigating the genetic basis for the evolution of complex systems over broad evolutionary timescales will inform us about conservation and the convergence of genes and gene function, with fundamental implications for our ability to predict gene function across species and at different scales of biological organization. There are different mechanisms by which complex trait evolution might occur. One model proposes that complex traits evolve mainly by the recruitment of homologous genes or existing genetic pathways [1]. Here, the genes and regulatory networks encoding complex traits are similar across organisms, in line with the idea of “deep homology”, where gene function is often conserved over vast evolutionary timescales [2]. In addition, traits may also evolve by co-opting genes from multiple existing genetic pathways and/or incorporating novel genes to form unique connections [3]. Here, rather different genes and gene networks may encode similar complex traits. These models are not mutually exclusive, and many traits could be encoded by a mix of homologous and nonhomologous genes. Identifying the extent of constraint and co-option and finding general rules to explain the variety of results are still major challenges in evolutionary biology.

Animal eyes are an excellent trait for investigating the evolution of complex systems, for several reasons. First, eyes are diverse and vary in complexity, ranging from lens eyes to compound eyes or simple eyespots. The evolutionary origin of eyes and photoreceptor cells has been a matter of ongoing debate aiming to determine whether they evolved multiple times independently or shared a common origin [4,5,6,7]. The evolution of eyes might have occurred independently in different animal groups at least 40–60 times, and some recent molecular studies suggest this number might be even higher [8,9]. Second, numerous studies have characterized many genes involved in eye development and phototransduction in model bilaterians [10,11,12]. These genes are used as a starting point to study the visual systems of other animals. Yet whether these models and candidate genes are predictive of biology and evolutionary trends in other organisms remains to be determined.

Here, we review the literature on the evolution of eyes, photoreceptors, and opsins, as well as opsin expression and proposed functions, including mechanisms of phototransduction across animals (Figure 1; Appendix A). A close look at the literature reveals diversity that is overshadowed not only by the ease of looking only for candidate genes across species but also by the traditional approach of classifying photoreceptor cells and opsins into ciliary cells employing “c-opsins” and rhabdomeric cells employing “r-opsins”. To emphasize the diversity that can underlie opsin-based light detection across animals, we highlight Cnidaria, by mining existing data to uncover potential genes involved in their phototransduction and the visual cycle. Studies on Cnidaria, as a sister group to Bilateria, inform us about the conservation and repeatability of genes encoding for eye development and function. In particular, we explore the expression of the homologs of phototransduction and visual cycle genes in two cnidarian species with published transcriptome data: *Hydra*, which lacks eyes but has a behavioral response to light, and *Tripedalia*, which has 24 eyes, including complex lens eyes. Overall, we find that the hypothesized phototransduction cascade of cnidopsin is distinct from the cascades of visual Gt and Gq-opsins (Figure 1). Furthermore, some cnidopsins may serve photoisomerase roles with unknown cascades [13].

## 2. Evolution of Eyes and Photoreceptor Cell Types

### 2.1. PRC Evolution

Photoreceptor cells (PRC) are photosensitive neurons, and the number of times that these visual structures have evolved is a matter for debate. Before the advent of molecular techniques, the main criterion to distinguish between PRCs was morphology [14]. Most PRCs use cell surface enlargements to employ large amounts of visual pigment rhodopsin to detect light [7,15,16,17,18]. Some PRCs have modified cilia, referred to as ciliary PRCs (Figure 2A). Others have microvilli, as in the arthropod rhabdom, leading to the name rhabdomeric PRCs (Figure 2B). Even though the distinction between ciliary and rhabdomeric PRCs is oversimplified, this differentiation has been widely used and led to different interpretations of their evolution [4,8,14,16]. Eakin suggested a common evolutionary origin with two main lines of photoreceptors: a ciliary line in cnidarians and deuterostomes and a rhabdomeric line in protostomes [19]. Vanfleteren suggested a ciliary origin for all photoreceptors and an inductive function of cilia in rhabdomeric PRCs [20]. In contrast, von Salvini-Plawen and Mayr (1977) rejected a common evolutionary origin and strongly promoted a polyphyletic origin of photoreceptors, because of structural and cellular differences [8]. Some recent studies have uncovered diversity obscured by a binary division of PRCs. For example, it is not unusual to find a cilium, or at least remnants of a cilium, in otherwise-rhabdomeric protostome eye PRCs [21] or PRCs with no cell surface enlargements at all [4]. Another example is that of cnidarians, which have photoreceptors with modified cilia but use cnidopsins for their transduction. Thus, to decipher the evolutionary history of PRCs, it will be important to understand these cell types in light of the cellular and molecular features that define their diversity and evolutionary history.

### 2.2. Genetics of Eye Development and PRC Determination

Understanding the genetics of eye evolution first focused on candidate genes in model bilaterians, which overshadowed the true diversity that we now understand. Early on, researchers identified a set of transcription factors (TF) that play important roles in eye determination and differentiation. Of particular importance was vertebrate *pax6* (homologous to *eyeless* and *twin of eyeless* in *Drosophila*), which can induce eye development in different body regions of *Drosophila* when misexpressed [7,22]. Even-more astonishing, the ectopic expression of squid or mouse *pax6* cDNA in *Drosophila* led to ectopic eyes morphologically similar to normal eyes [7,22]. These results put new emphasis on the idea of a monophyletic origin of eyes under the control of the “master control gene” *pax6*. Later, scientists discovered that many conserved TF genes are critically involved in eye development, including *dachshund*, *eyes absent*, *six1/2*, *six3/6*, and *otx* [23]. Similar to *pax6*, these TFs induced ectopic eyes when misexpressed and induced absence of eyes when knocked down. The idea of a monophyletic origin of eyes appeared to be a reasonable assumption based on homologous genes involved in the eye development of different animal groups. However, a common evolutionary origin is not the only possible explanation, in that these similarities could also result from co-option events (i.e., eyes evolved independently by recruiting similar developmental genes) [4,5,6,24,25].

Recent findings highlight an increasing diversity of visual-system genetics and some plasticity in the genes and networks that carry out vision-related cell fate and development. Even though studies in non-model organisms do reveal that some of the same gene families are used in eye development, many animal groups use paralogous genes, different combinations of genes, or different regulatory networks. For example, members of the pax, rax, and six transcription factors are necessary for eye development, but the members of these gene families with a role in cell fate vary [9]. In addition to different genes’ developing eyes themselves, PRC fate can also be driven by variation in gene regulation. In vertebrates, *crx*, *otx2*, and *rax* are expressed in all PRCs, but different factors control the opsins expressed in rods (nrl, nr2e3, nr1d1, pias 3, and fiz1) compared with cones (thbr and rxrg for Lws opsin; tbx2 for Sws1) [26]. In insects, a set of genes (*senseless*, *prospero*, *spalt*, and *spineless*) is used for PRC specification, but variation in gene regulatory networks drive differences across a compound eye and between species [27,28,29].

### 2.3. Cnidarian Photoreception

Outside of Bilaterians, the evolution of eyes probably occurred convergently in different cnidarian phyla at least nine times, and these independent origins may use different genes for eye development [30,31]. For a review of cnidarian visual systems, light detection, and behavioral responses to light, see Birch et al. [32]. In brief, cnidarians can possess extraocular ciliary photoreceptors and eyes of varying degrees of complexity. The latter includes so-called ocelli, a collection of photoreceptor cells associated with pigment cells and also includes more-complicated eyes with retinas, corneas, and lenses. Photoreceptors and eyes in cnidarians are often found in structures called rhopalia. Rhopalia are sensory structures found in cubozoans and scyphozoans that can contain condensed neurons, gravity receptors, photoreceptors, and sometimes species-specific numbers of low-resolution lens eyes [32,33]. In cnidarians that lack rhopalia, eyes are found in modified structures associated with tentacles, such as tentacle bulbs in Hydrozoa and rhopalioids in Staurozoa [30,31]. The genetics underlying convergently evolved cnidarian eyes are still not well understood, but studies have characterized the expression of *eya*, *six*, and *pax* genes in some species. In the hydrozoan *Cladonema radiatum*, *eya* is expressed in eyes and gonads [34]. Both *six1/2* and *six3/6* are expressed in the eyes of *Cladonema*, and this expression increases during eye regeneration [35]. Different members of the *pax* gene family also play a role in eye formation. For example, the cubozoan *Tripedalia* may use *paxB* for eye formation, while the Hydrozoan *Cladonema* uses *paxA* [36]. A more recent study comparing the transcriptomics of convergently cnidarian evolved eyes found variation in gene expression underlying eyes of unique origins [37]. Moreover, in Cnidaria, opsins have undergone species-specific duplications and expansions [38,39] such that investigating the opsins and cascades used by these species will tell us more about the visual system diversity in this group.

### 2.4. Opsins and Phototransduction Cascades

Although the main functions of PRCs are the detection and induction of a light response, they often employ different proteins to fulfill these functions (Figure 1). Phototransduction is initiated by rhodopsin, an opsin protein coupled to a vitamin A-derived chromophore molecule, usually retinal [7,11,40]. When the chromophore absorbs light, it changes from an 11-*cis* to an all-*trans* conformation, activating the opsin [41,42]. Opsins belong to a large family of G-protein coupled receptors, with a complex taxonomy based on different considerations, including the morphology of cells where they are expressed, the signal cascade that they initiate, expression pattern, taxonomic scope, or their phylogenetic relationships [12,40,41]. Opsins are phylogenetically split into several (broadly accepted) major families: c-opsins, r-opsins, xenopsins, cnidopsin, anthozoan specific opsins, and tetraopsins (Figure 1) [30,43,44,45,46,47]. However, the relationships within and among some of these groups remain unresolved [43,44,45,46,48,49]. Depending on the opsin family, a different G-protein is activated, initiating different signaling cascades that result either in the hyperpolarization or in the depolarization of the PRC [12,50]. More precisely, the Gα-subunit of the G-protein dissociates from the βγ-complex and acts as an effector for other enzymes of the cascade [42]. As a consequence, it is mainly the Gα-subunit that is referred to in studies on G-protein and opsin interactions. In this section, we provide an overview of the major opsin families and their phototransduction components in more detail.

### 2.5. Gi/Gt-Opsins (Traditionally C-Opsins)

Visual c-opsins are the main visual opsin in vertebrate eyes, but phylogenetically related opsins are also present in brains of vertebrates and protostomes [40,51,52]. These opsins are found in ciliary cells and function via the G-protein subunit Gi (Transducin (Gt) in vertebrates).

#### 2.5.1. Gi/Transducin Cascade

Phototransduction in rod cells of vertebrates is well studied and used as a model for the c-opsin cascade (Figure 1 and Figure 2). Rhodopsin activates the Gα subunit of Transducin (G-protein), which in turn activates a cGMP Phosphodiesterase (PDE). PDE hydrolyzes cGMP, leading to a decrease in the cellular concentration of cGMP. The reduction of cGMP leads to the closure of cyclic nucleotide–gated channels (CNGs), preventing an influx of positively charged Na^+^ and Ca^2+^ ions. This reduces the electrical current of the PRC, and by stopping an otherwise-constant inward current, the PRC hyperpolarizes [12,50,53]. In order to end signaling, rhodopsin is phosphorylated by rhodopsin kinase, followed by the binding of arrestin [54].

#### 2.5.2. Deuterostome Gi/Gt-Opsins

The most recognizable members of the Gi/Gt-opsins are the visual opsins of vertebrates expressed in rods and cones of the retina. These are by far the best-studied opsins, especially bovine rhodopsin. In different animal groups, such as fish, this group is further subdivided into at least five subgroups, depending on spectral sensitivity: LWS (long-wavelength sensitive) detects yellow/red, SWS1 (short-wavelength sensitive 1) detects UV, SWS2 (short-wavelength sensitive 2) detects blue, RH2 (rhodopsin-like) detects green, and for RH1 (rhodopsin), the rod opsin has peak sensitivity in the blue/green [55]. These visual opsins have undergone multiple events of duplication, loss, and divergence [56,57]. The expression of visual opsins also contributes to the divergent and convergent evolution of color vision [58].

Closely related to vertebrate opsins used for color vision are vertebrate opsins, traditionally referred to as nonvisual, classically described as “c-opsins” even though their transduction cascades remain largely unknown. This group includes pinopsins, expressed in the pineal organ of birds, reptiles, and amphibians [59,60,61], and has recently been found in the retina of nonteleost fish and frogs [62]. Pinopsin is expressed in rod cells of the spotted gar and in rod and cone cells of the western clawed frog, with peak absorbance at ~470 nm, and may function in dim-light detection [62]. Parapinopsin is another nonvisual opsin expressed in the pineal and parapineal organs of fish and amphibians [63,64]. Teleost fish have two copies of parapinopsin: one sensitive to UV, likely involved in wavelength discrimination, and the other sensitive to blue, hypothesized to function in melatonin secretion [65]. Pinopsins and parapinopsin function via a Gt protein [66,67], but other components of their signaling cascade have yet to be described.

Vertebrate ancient-opsins (VA-opsins), encephalopsins, and TMT-opsins also belong to the nonvisual “c-opsin” group. VA-opsins are expressed in the bird hypothalamus and fish retinas and brains [68,69,70,71]. Encephalopsins are found in mice and humans and are broadly expressed in different tissues, primarily the brain, but also the heart, lung, liver, kidneys, testes, and retina [42,72,73,74,75]. Despite broad expression, the function of encephalopsins remains unknown, with suggested functions in circadian rhythm or melatonin production [75]. TMT-opsins, or teleost multiple-tissue opsins, are expressed in neuronal and non-neuronal tissue in fish and may be associated with the adjustment of the inner circadian clock [42,76]. TMT-opsins can be divided into three groups: TMT1, TMT2, and TMT3. Both TMT1 and TMT2 are sensitive to blue light, bind all-*trans* retinal, and function through the Gi and Go cascades [77,78]. TMT1 can also retain an 11-cis, 7-cis, and 9-cis chromophore [78]. Work in the medaka fish shows that mutations in TMT1 result in increased avoidance behavior and increased average activity [79].

Although most research has focused on vertebrates, they are not the only deuterostomes that have ciliary opsins. Sequences that group with traditional vertebrate c-opsins in phylogenetic analyses have been found in *Branchiostoma*, the ascidian *Ciona intestinalis*, several echinoderm species, and hemichordates [74,80,81,82]. In fact, in *Branchiostoma*, the “c-opsins” are expressed in the so-called frontal eye [81], and in *C. intestinalis*, “c-opsin” is expressed in the ocellus [80].

#### 2.5.3. Protostome Gi Opsins

The ciliary and rhabdomeric PRC dichotomy led early researchers to believe that “c-opsins” were found only in deuterostomes and r-opsins in protostomes; thus, the discovery of a c-opsin in a protostome was a breakthrough in photoreceptor research. The first ciliary opsin found in a nonvertebrate animal was the “c-opsin” of the annelid *Platynereis dumerilii* expressed in the brain [51]. These opsins transduce signals using Gi/Go and may function in circadian photoentrainment [51,83] and in larvae as a depth gauge [84]. Closely following the discovery of the *Platynereis* “c-opsin”, pteropsin was found in the honeybee brain [52]. Pteropsin is expressed in the central nervous system and brain (not eyes) of other arthropods and may play a role in circadian clocks [52,85,86]. The only reported exception to date from a pattern of protostome “c-opsins” expressed only outside eyes is the PCR amplification of a c-opsin from the brain and eye of an onychophoran [85]. However, signal strength is weak in the eye sample, and the results have not yet been confirmed by qPCR, in situ hybridization, or antibody staining. The phototransduction cascade by which pteropsins signal is completely unknown.

### 2.6. Gq-Opsins (Traditional R-Opsins)

Traditional r-opsins are found in the main cells of the visual system in protostome eyes (microvillar PRCs) but can also be found in nonvisual tissues and in deuterostomes. Where studied, r-opsins activate a Gq phototransduction cascade.

#### 2.6.1. Gq Cascade

The best-studied phototransduction process in any invertebrate employing rhabdomeric opsins is from *Drosophila*, which often serves as the model for r-opsin cascades (Figure 1 and Figure 2). Upon light detection, rhodopsin changes to its active conformation (metarhodopsin), which activates Gq. Gq then activates the enzyme phospholipase C (PLC) that interacts with a membrane phospholipid (PIP_2_), generating diacylglycerol (DAG) and inositol 1,4,5-tris-phosphate (IP_3_). The exact function of DAG and IP_3_ still needs to be determined, but two classes of transient receptor potential channels (TRPC and TRPL) open. The activation of TRP and TRL may also be driven by PIP2 cleavage, changing the cell surface area and mechanically opening the channels (for a TRP opening review, see Hardie and Juusola 2015 [10]). The opening of these channels allows an influx of positively charged Ca^2+^ ions, leading to a depolarization of the PRC membrane. Termination happens when rhodopsin has been deactivated [10,50,87].

#### 2.6.2. Protostome Gq-Opsins

Protostome visual Gq-opsins (traditionally “r-opsins” or “canonical r-opsins”) can be subdivided into at least three subgroups, depending on spectral sensitivity: short-wavelength or ultraviolet (UVRh); medium wavelength, sometimes referred to as blue (BRh); and long wavelength (LWRh) [88]. Gq-opsins can also be found in nonocular tissues, such as the brain, ventral body region, skin, or bioluminescent organs of cephalopods [86,89,90,91]. Accordingly, their main function in protostomes is vision, but they can also be used for other light-dependent functions, such as circadian clocks, photo re-entrainment, and phototaxis [86,92,93,94]. Moreover, these opsins can also have light-independent functions, such as sensing heat, taste, touch, and sound [95]. A closely related group to the visual Gq-opsins was originally described in arthropods and named “arthropsins” but has since been found in lophotrochozoans and a few cephalochordates [45,85,96]. The signaling cascade of arthropsins remains unknown.

#### 2.6.3. Deuterostome Gq-Opsins

Similarly surprising to the finding of “c-opsins” in protostomes was the discovery of an “r-opsin” homolog in vertebrates, named “melanopsin” [97,98]. Melanopsin is expressed in the intrinsic photosensitive retinal ganglion cells of vertebrate eyes but is not involved in image-forming vision [97]. Melanopsin is likely involved in circadian photoentrainment and functions through Gq transduction similar to protostome r-opsins, although isolated melanopsin can also activate transducin (Gi) [99,100]. Several other r-opsin homologs in addition to melanopsin are found in different deuterostomes, including in *Branchiostoma* [101] and echinoderms [82,102].

## 3. Tetraopsins

Tetraopsins, or Group 4 opsins, actually comprise three distinct subgroups: Go-opsins, neuropsins, and the photoisomerase subfamily [40,43,45].

### 3.1. Go-Opsins

Go-opsin was first described in ciliary PRCs of the scallop *Patinopecten yessoensis*, so named because it activates G-alpha-o [103]. Go-opsin was also found in echinoderms, *Branchiostoma*, and *P. dumerilii* [82,104,105,106,107,108]. In *P. dumerilii* larval eyes, Go-opsin is coexpressed with an r-opsin, and knocking it down reduces phototaxis to cyan light [107]. 

#### Go Cascade

Phototransduction for Go-opsins leads to hyperpolarization similar to Gi/Gt-opsins, but the former uses a different signaling cascade. The Go subunit activates a guanylate cyclase (GC), resulting in an increase in cGMP. This opens CNG channels that allow an influx of K^+^ ions, resulting in hyperpolarization [12,42,103].

### 3.2. Neuropsins

Neuropsins (OPN5 in vertebrates) exist in both deuterostomes and protostomes but are poorly characterized. In vertebrates, neuropsins are expressed in the spinal cord, testis, brain, and eye [109,110], but it may be mainly a deep brain photoreceptor with the ability to detect UV light and interact with Gi [111,112,113]. In the marine annelid, *Capitella teleta*, neuropsins are expressed in brain regions, but their function remains unknown [114]. Transgenic studies in mice reveal that OPN5 plays a role in skin circadian clocks and light sensitivity [115].

### 3.3. Photoisomerases

Three opsin groups are summarized as the retinal photoisomerase subfamily and named according to their close phylogenetic relationship, even though not all the genes necessarily function as photoisomerases, including retinal G-protein-coupled receptor (RGR), peropsins, and retinochromes [42,43]. Opsin photoisomerases are opsin proteins that may function in converting the activated all-*trans* retinal back into its reactive conformation of 11-*cis*. RGRs are found in vertebrates, retinochromes in invertebrates, and peropsin in both. Presumably, these opsins share the ability of light detection and retinal photoisomerization, and all of them preferentially bind retinal in all-*trans* conformations, even though they can also bind 11-*cis* conformations in vitro [105,116,117]. However, one of the key functions of opsins, the activation of G-proteins, seems at least partly lost in this opsin group. Although the characteristic lysine residue K296 is present, some of the important amino acids for G-protein activation, such as the tripeptide and the NPxxY motif, deviate in retinochromes and RGRs but are largely conserved in peropsins [42,118]. As a consequence, retinochrome and RGR are most likely not capable of initiating phototransduction, whereas peropsins at least preserved the potential capability of doing so. Where incapable of phototransduction, their main function seems to be photoisomerization [118]. Another recent study described a group of opsins related to RGRs and retinochromes, in which the characteristic lysine group has been replaced by glutamic acid, leading to the name gluopsins. Because of their potential inability to bind retinal because of the lack of lysine, their function is at present in question [119].

#### 3.3.1. RGRs

RGR was first found in the vertebrate retinal pigment epithelium and predicted to have sensitivity to blue and near-UV wavelengths [116]. RGRs are found in several vertebrate and invertebrate species [88,120,121]. Mutations of *RGR* in mice decrease 11-*cis* retinal and opsin activity and increase accumulation for all-*trans* retinal [120]. Similarly, in retinal ganglion cell cultures, the knockdown of *RGR* results in significant differences between 11-*cis* and all-*trans* retinal in both light- and dark—adapted conditions [121].

#### 3.3.2. Peropsins

Peropsins are found in both vertebrates (chicken and mouse) and invertebrates (*Branchiostoma*, *Platynereis*, and jumping spider) [105,122,123,124]. The expression of peropsin was known only from nonvisual PRCs. For instance, it is present in the pineal organ of chickens [125], the brain of *P. dumerilii* [124], nonvisual cells of the spider retina [126], and cell-surrounding photoreceptors in the horseshoe crab [127]. Recently, however, peropsin was found to be highly expressed in starfish eyes, but whether expression is in the photoreceptor or supporting cells is yet unknown [102]. Similar to RGR, mutations of peropsin in mice result in the increased accumulation of all-*trans* retinal, suggesting a role for peropsin in the light-dependent regulation of retinal [128].

#### 3.3.3. Retinochromes

Retinochrome was first described in cephalopods and functions in chromophore photoisomerization [129,130,131]. Retinochrome sequences are found in multiple protostome species, including gastropod and chiton mollusks, brachiopods, *P. dumerilii*, and arthropods [45]. Retinochrome is expressed in the larval eyes of the chiton *Leptochiton asellus* [118], and a retinochrome-like gene is expressed in the eye pigment cells of the butterfly *Heliconius melpomene* [132]. Yet the role and function of retinochrome outside of mollusks have not been functionally demonstrated.

## 4. Xenopsin

Xenopsins, a recently described type of opsin, have been changing the conversation of PRC evolution because they have been found coexpressed with r-opsins and c-opsins [46,133,134]. Xenopsin, originally suggested to be a c-opsin, was first reported as expressed in the larval eye of *Terebratalia transversa* [135]. Phylogenetic analyses have constantly found that these opsins form a clade separate from c-opsins [45,46,133,134]. Xenopsins exist in Lophotrochozoa and in Cnidaria as cnidopins (see below) and therefore were present before the Cnidaria and Bilateria split [45]. In larvae of *Leptochiton asellus*, xenopsin and r-opsin are coexpressed in PRCs with both cilia and microvilli [46]. In the larvae of tiger flatworm *Maritigrella crozieri*, both xenopsin and r-opsin are expressed in eyes, with xenopsin localized to cilia [133]. Xenopsin is also present in the larvae of the annelid *Malacoceros fuliginosus* and Bryozoan *Tricellaria inopinata* localized to cilia [134]. *Malacoceros* also has a c-opsin, indicating for the first time that xenopsin and c-opsin can occur together in a single genome [134]. The G-protein by which xenopsins function remains unknown.

## 5. Cnidopsin

The first cnidopsins (originally named cnidops) were discovered by analyses of genomic data of cnidarians [136]. Several studies have confirmed that many of these opsins form a phylogenetic clade on their own [30,38,43,45,47]. Phylogenetically, these opsins appear to be a sister group to xenopsins (Figure 1) [45,46]. Cnidopsins are quite multifaceted in terms of expression, with some being eye specific, whereas others are expressed in different tissues, such as gonads, tentacles, manubrium, or umbrella [38,39,137]. 

### 5.1. Gs Cascade

Where known, cnidopsins employ a Gs cascade [137,138]. In the box jellyfish *Carybdea rastonii*, Gs is colocalized with opsin in the outer segment of the cubozoan eye and responds to light by activating adenylyl cyclase [138]. The activation of Gs leads to a decrease in the cellular cAMP concentration [138]. The Gs pathway is also activated by two opsins upregulated in the rhopalia of the box jellyfish *Tripedalia*, but the signaling pathways of the other *Tripedalia* opsins remain unknown [137]. In *Hydra,* opsin *HmOps2* is coexpressed with *CNG* and *Arrestin*, and on the basis of pharmacological knockdown, *CNG* is necessary for a response to light [139,140].

### 5.2. Anthozoa Opsins

In addition to cnidopsins, cnidarians in the group Anthozoa (sea anemones and corals) have two additional groups of opsins, called anthozoa opsins I and anthozoa opsin II (ASO-I and ASO-II; Figure 1) [44]. These opsins are divided into subtypes: ASO-I into subtypes 1 and 2 and ASO-II into subtypes I, 2.1, and 2.2 [30,47]. In the sea anemone *Exaiptasia diaphana*, the expression of some of these ASOs varied between larval and adult stages and between symbiotic and aposymbiotic individuals [47]. Anthozoans are also unique among cnidarians in using cryptochrome to detect light [47]. Preliminary work suggests Anthozoans may possess multiple phototransduction cascades because heterologously expressed opsins from the coral *Acropora palmata* might initiate a Gq and “Gc” (c for cnidarian) cascade [141].

### 5.3. Expression of Bilaterian Phototransduction Genes in Cnidarians

To investigate which genes may function in phototransduction across cnidarians, we used bilaterian genes as candidates for functions in light detection. We hypothesized that genes used for light detection would be upregulated in light-sensitive tissues in cnidarians (rhopalia of *Tripedalia* and *Hydra* heads; Figure 3).

#### 5.3.1. Methods

Peptide fasta files were obtained for the genomes of *Danio rerio*, *Gallus gallus*, *Drosophila melanogaster*, *Mus musculus*, and *Homo sapiens* from Ensembl. The peptide sequences for *Nematostella vectensis* were obtained from Ensembl Metazoa. The *Aurelia* genome and gene models were obtained from Gold et al., 2019; for *Clytia hemisphaerica*, from Leclère et al., 2019; and for *Nemopilema nomurai*, from Kim et al., 2018. Protein models for *Hydra vulgaris* were obtained from the *Hydra* genome browser made available by the NHGRI. *H. symbiolongicarpus* and *H. echinata* draft transcriptomes were also obtained from NHGRI. The draft *Sarsia tubulosa* transcriptome assembly was obtained from [37]. Transcriptome assemblies for *Acropora millepora*, *Alatina alata*, *Aurelia aurita*, *Calvadosia cruxmelitensis*, *Chironex yamaguchii*, *Chrysaora quinquecirrha*, *Clavularia* sp., *Copula sivickisi*, *Craterolophus convolvulus*, *Dynamena pumila*, *Haliclystus auricula*, *Haliclystus sanjuanensis*, *Lucernaria quadricornis*, *Millepora alcicornis*, *Millepora complanata*, *Millepora squarrosa*, *Morbakka virulenta*, *Protopalythoa variabilis*, *Physalia physalis*, *Podocoryna carnea*, *Porpita porpita*, *Rhopilema esculentum*, *Tripedalia cystophora*, *Turritopsis* sp., *Velella velella*, and *Xenia* sp. were downloaded from NCBI accessions and authors, as listed in Appendix A.

For all transcriptome assemblies, we used TransDecoder v. 5.5.0 to find the peptide sequence for the longest open-reading frame using TransDecoder.LongOrfs. Peptide fasta files for all 39 species were moved into a single directory. Cd-hit v. 4.8.1 was used to remove probable isoforms from all fasta files using parameters -c 0.9 and -n 5 [142,143]. Orthofinder v. 2.4.0 was then used to detect orthogroups between species by using a blast e-value of 1 × 10^−5^ [144]. We used a gene list from [145,146] to look for candidate genes involved in cnidarian phototransduction. We extracted sequences for some of these genes from FlyBase for *Drosophila melanogaster* and from NCBI for *Hydra vulgaris*. Extracted sequences were aligned to the *Tripedalia* assembly peptide sequences by using command line blast. We used reciprocal blast on NCBI GenBank to verify candidate gene annotation. We then identified the orthogroups for these genes. Sequences for candidates were aligned using MAFFT v. 7.475, and phylogenetic trees were calculated using iqtree v. 2.0.3 with parameters -m MFP -B 1000 -alrt 1000 -T 8 [147]. After a visual inspection of the phylogenetic trees, some were repeated with only proteins that had sequence length > 150, 200, or 300 aa. Trees were visualized and edited in iTOL v. 6.1.1 [148]. Using these trees, we characterized potential gene duplications or losses in *Hydra*, *Aurelia*, and *Tripedalia* because they belonged to different subphylums and because *Aurelia* and *Tripedalia* have independently evolved eyes. Expression was plotted for *Hydra* and *Tripedalia* because the RNA-seq for different tissues is publicly available. *Hydra* RNA-seq data were downloaded from GEO accession GSE127279 [149] and *Tripedalia* from NCBI SRR8101518-SRR810526 [150]. For both species, reads were mapped to the reference assembly by using bowtie2 v. 2.4.1 [151] and RSEM v. 1.3.3 [152].

#### 5.3.2. Results

To investigate which genes may function in phototransduction across cnidarians with independently evolved eyes, we used bilaterian genes as candidates for functions in light detection. We hypothesized that genes used for light detection would be upregulated in jellyfishes’ light-sensitive tissues (*Hydra* head and *Tripedalia* rhopalia; Figure 3). Because of their central importance in defining phototransduction cascades, we first looked for G-alpha genes in cnidarian genomes and transcriptomes to identify homologs of Gs, Gi, Gq, and Go (Table 1; Figure 3 and Appendix A). We identified a potential duplication of Gi in *Aurelia* and two Gs-like genes in species excluding *Hydra* (Table 1; Appendix A). Gs, the phototransduction signaling protein in box jelly eyes, was in fact expressed at higher levels in *Tripedalia* rhopalia relative to other tissues. In *Hydra*, Gs was expressed at low levels in all tissues (Table 1; Appendix A). The next component that we surveyed was the G-protein subunit beta (GNB). We found two groups of GNB genes, *GNB5* orthologs and *GNB1-4-like* genes (Appendix A). We identified three copies of *GNB1-4-like* in *Hydra*, but two of them lacked or had low expression (Table 1; Appendix A). *GNB1-4-like* was expressed at higher levels in *Tripedalia* compared with *GNB5* and had higher expression in the rhopalia compared with other tissues (Appendix A).

Two classes of ion channels are known in bilaterian phototransduction: TRP/TRPL used by *Drosophila* and CNG by vertebrates (Figure 2). In cnidarians, *TRPC* was present in all species with a potential duplication in *Aurelia* (Table 1; Appendix A). *TRP* had higher expression in the *Tripedalia* rhopalia (Appendix A). For *CNG*, a *CNG-like* gene was present in all species with two paralogs in *Aurelia* (Table 1; Appendix A).

The enzyme proposed to function in cnidarian phototransduction is adenylyl cyclase (AC). We identified orthologs for three types of AC enzymes in cnidarians (AC-type2, AC-type5, and AC-type9) (Table 1; Appendix A). Only *AC-type9* had a duplication in *Hydra* (Table 1; Appendix A). *AC-type2* and *AC-type9* had higher expression in the *Tripedalia* rhopalia (Appendix A). GC is the enzyme used in Gt opsin phototransduction, we found three GC genes in cnidarians (*GC-alpha*, *GC-beta*, and *GC88E*) (Table 1; Appendix A). We found two assembled transcripts for *GC-alpha* in *Tripedalia. GC88E* had three copies in *Aurelia*, two in *Tripedalia*, and was absent in *Hydra* (Table 1; Appendix A). In terms of expression, *GC-beta* had higher expression in *Hydra* tentacles, and four GC genes had higher expression in the *Tripedalia* rhopalia (Appendix A). For PLC, used by Gq-opsins, we identified subunits beta1 and beta4 (Table 1; Appendix A). *Hydra* has three copies of *PLC-B1* in tandem but one copy with no expression, and *PLC-B1* had higher expression in *Tripedalia* rhopalia. For PDE, we did not find an ortholog of PDE6 (the gene used by vertebrates), but we found a closely related group (Appendix A). We also identified *PDE5A* and *PDE11A* genes in cnidarians (Appendix A). These genes, for the most part, had similar expressions in all tissues (Appendix A). G-protein signaling is terminated by activated rhodopsin binding arrestin or being phosphorylated by rhodopsin kinase (Rhk). In cnidarians, we found one copy of *arrestin* and *Rhk* in all species (Appendix A). Both these genes had higher expression in *Hydra* tentacles and *Tripedalia* rhopalia (Appendix A).

According to our observations, it seems as though there may be differences in the phototransduction cascades even within Cnidaria. We predicted that genes involved in phototransduction would be more highly expressed in light-sensitive *Hydra* and *Tripedalia* tissues. As predicted, there are instances where orthologs were highly expressed in *Tripedalia* rhopalia and *Hydra* heads, implying similarity in cascades. However, there were other gene families where different paralogs were upregulated in the two species or there was no difference in expression across tissues. These results indicate that some cnidarian species may co-op paralogs or different genes in their phototransduction cascades. However, we cannot conclude without additional functional tests that the genes more highly expressed in the eye-bearing tissues have a direct role in phototransduction. The rhopalia are highly sensory tissues, so it could be that some of the G-protein components function in modulating other sensory modalities. Another caveat of this preliminary research is that without genomes for *Tripedalia*, we cannot confidently conclude whether genes are duplicated or instead represent isoforms when we identify multiple transcripts. Additionally, the lack of replication for *Tripedalia* and limited tissue sampling and replication in other species may bias expression results. For validation of the molecular evolution, expression, and function of these genes we need to (1) assemble genomes for more cnidarian species, (2) increase replication and tissue sampling, and (3) perform functional validations of genes highly expressed in eye-bearing tissues.

## 6. Placopsins and Ctenophore Opsins

Placopsins, first reported in the *Trichoplax* genome [153], and a ctenophore opsin (*Mnemiopsis leidyi* 3) are proposed to be an outgroup to other animal opsins [48,146,154]. *Mnemiopsis leidyi* possesses three opsins that do not form a phylogenetic group, but their relationship to other opsins varies under different models and with different outgroups [146,154]. While homologs of traditional c-opsin and r-opsin cascades are expressed in the *M. leidyi*, their role in phototransduction or other sensory modalities has not been validated [146].

## 7. Visual Cycle

In addition to different signaling cascades, opsins vary in the mechanisms used to recycle receptive rhodopsin, termed “the visual cycle”. One main difference between visual Gq and Gi/Gt-opsins is that the retinal chromophore remains bound to Gq-opsins after absorbing light. Still bound to the Gq-opsin, retinal can absorb light of a different wavelength and change back to the 11-*cis* conformation. For visual Gt-opsins, on the other hand, retinal dissociates after the conformational change to all-*trans*. For this reason, Gt-opsins are referred to as monostable pigments, whereas Gq-opsins are called bistable pigments [155]. Yet both visual systems have additional mechanisms to recycle retinal back to the 11-*cis* conformation.

### 7.1. Vertebrate Visual Cycle

Vertebrates have light and dark visual cycles, and different visual cycles for their rod and cone PRCs [156,157,158]. In the dark visual cycle of the rod PRCs, all-*trans* retinal is reduced to all-*trans* retinol by retinol dehydrogenases (RDHs), specifically RDH8 (and potentially additional RDHs) [159]. All-*trans* retinol is transported through the interphotoreceptor matrix by chaperoning interphotoreceptor retinoid-binding protein (IRBP or RBP3) into the adjacent cells of the retinal pigment epithelium (RPE), where it is bound by the cellular retinol binding protein, CRBP1. While still bound to CRBP1 all-*trans* retinol is esterified by the lecithin retinol acyltransferase (LRAT) to all-*trans* retinyl esters (e.g., all-*trans* retinyl palmitate). Consecutively, the isomerohydrolase RPE65 hydrolyzes and isomerizes the all-*trans* retinyl ester to the respective fatty acid and 11-*cis* retinol, which is then bound by the retinaldehyde binding protein RLBP1 (CRALBP). Subsequently, RDH5 catalyzes final oxidation to 11-cis-retinal before retinal is transported back to the rod PRC by IRBP to complete the cycle and regenerate rhodopsin [157,158,159,160,161,162].

In contrast to rods (used in dim light) that receive all their 11-*cis* retinal exclusively from the RPE, cone PRCs (used in bright light) receive 11-*cis* retinal from RPE and Müller cells. These Müller cells are capable of an alternative visual cycle, referred to as cone visual cycle. In mice, Müller cells require light, RGR, and RDH10 to recycle retinal [163]. Before RGR’s function in Müller cells was discovered, RGR’s photoisomerase function in the light visual cycle was predicted in the RPE [54,164]. In the RPE, RGR may function together with RDH5 and RPE65, but the cycle remains to be understood in its entirety [158,165,166]. The exact function of this opsin has yet to be determined, but a key role in rhodopsin regeneration under constant illumination, a storage for retinoids, and regulation of LRAT has been suggested [161,167].

### 7.2. Invertebrate Visual Cycle

Because of the bistability of visual Gq-opsins (traditionally r-opsins), it was assumed that invertebrates do not require visual cycling. However, it seems that the photoizomerization of metarhodopsin alone could not recover the high amount of receptive rhodopsin. Instead, there must be another mechanism that ensures a constant supply of 11-*cis* retinal [131]. The mollusk group of cephalopods has been the focus of this research for several decades and delivered first insights into invertebrate retinal recycling at a molecular level. In the eye PRCs of cephalopods, a second seven transmembrane protein with great similarity to opsins was discovered [129,131]. This opsin preferentially bound retinal in the all-*trans* conformation and released it in the 11-*cis* conformation (Figure 4). It was given the name retinochrome, a sort of reverse opsin [129,131,157,168,169]. In squid, retinochrome is expressed in the inner segments of the PRCs and the basal regions of the outer segments, which is a contrast to r-opsins, which is only found in the rhabdomeric microvilli [131,169,170]. Therefore, retinal transport is carried out by retinal binding protein (RALBP) that binds retinal in both conformations, 11-*cis* and all-*trans* [118,130,131,171]. 

For a long time, retinochrome was only known in cephalopod eyes, and the rhodopsin-retinochrome system of cephalopods was the only retinal-reisomerization process (in addition to photoisomerization by bistable pigments) known in invertebrate species. However, retinochrome has now been found in the larval eyes of two additional mollusks: the chiton *L. asellus* and the gastropod *Conomurex luhuanus* [118,171]. In chiton eyes, retinochrome is coexpressed with homologs to RLBP1, r-opsin, RALBP, and multiple retinal dehydrogenases [118]. The chiton eyes also express xenopsin, whose sequence resembles a c-opsin in terms of its monostability, which might require a vertebrate-like visual cycle. Accordingly, the visual cycle in squids and chitons might be more complex than previously proposed.

## 8. Cnidaria Visual Cycle

Cubozoan opsins, similar to vertebrate opsins, undergo bleaching, suggesting presence of a visual cycle that has yet to be described [172]. There is no cnidarian opsin that groups closely with known photoisomerases RGR or retinochrome [30,118]. It could be that a cnidopsin gene is co-opted as a photoisomerase, but that remains to be validated. In *Tripedalia*, an antibody against a cnidopsin detected protein in all eye types, suggesting a function as a photoisomerase [13]. Although specific photoisomerase genes are yet unproven in cnidarians, cnidarians possess homologs of genes used in other animals for chromophore transport. We found two members of the *SEC14-like* family (protein 2 and protein 5) that have similar expression in all tissues (Table 1; Appendix A). In addition to finding homologs of *clavesin/RLBP1* (Appendix A), we also found a cnidarian-specific gene family whose top BLAST results match *RLBP1*, so we refer to these genes as *RLBP1-like* (Appendix A). These genes are expressed at higher levels in the rhopalia of *Aurelia* and *Tripedalia* and of *RLBP1-like* in the tentacles of *Hydra* and *Sarsia* (Appendix A). *RLBP1-like* was detected in cnidarians, in a recent paper, but the amino acids necessary in other animals for retinal binding were lacking, so it may not be involved in the visual cycle [118]. For RDHs, we found a gene family matching aldehyde dehydrogenase (ALDH). Based on transcriptomes, this gene family may have duplications in *Aurelia* and *Tripedalia* (Appendix A). These genes were lowly expressed in *Hydra* and similarly expressed across tissues in the other species (Appendix A).

## 9. Conclusions and Future Directions

Eyes are complex traits that require multiple independent components coming together to function and evolve. Across animals, eyes can vary in structural complexity and probably evolved convergently multiple times. Early studies of eye and PRC evolution focused on comparisons between *Drosophila* and mammals, overshadowing much diversity. Recent studies and our work in cnidarians have begun to explore these traits outside of model bilaterians to find that eyes can function using a much wider variety of components. By synthesizing what is known about the cascades of different opsins, it becomes clear that there are many unknowns (Figure 1). Photoreceptor structures, opsin phylogenetic relationships, and opsin expression patterns hint at functional groupings, but functional studies into their regulatory networks and phototransduction cascades are necessary to determine opsin function and evolution. Our data, presented in this study, highlight the diversity of eye-specific proteins within different cnidarian species. Specifically, the presence of phototransduction elements that are associated with different opsin groups in other animals hints at a general pattern of diverse solutions to similar problems during evolution. Generally, this appears to corroborate the idea of the convergent evolution of eyes among animal phyla and specifically within cnidarians. The discussions and studies of eyes, PRCs, and opsin evolution now needs to move away from the traditional oversimplified distinctions and candidate gene approaches to incorporate the diversity being uncovered. Future work should focus on genome-wide approaches in non-model organisms to uncover species-specific and additional genes that may be co-opted for eye and visual functions. Advances in technologies for genome manipulation will allow researchers to test the functions of eye development and phototransduction genes in model and non-model organisms to further validate the extent of the conservation and diversity of these traits. Applying what we learned to complex traits more generally, visual research tells us that seemingly similar traits can have a deep diversity of underlying components.

## Figures and Tables

**Figure 1 cells-11-03966-f001:**
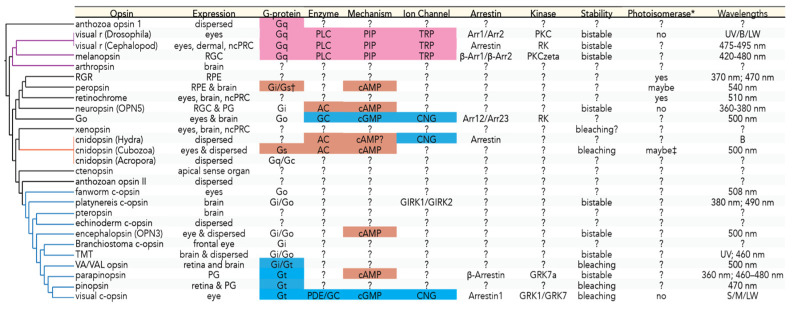
Diversity of opsins and phototransduction cascades. **Left**: Opsin phylogeny with purple branches for traditional r-opsins, blue branches for c-opsins, orange branches for cnidopsins. **Right**: Table with opsin characteristics. Typical components of r-opsins are in pink boxes, of c-opsin in blue, and of cnidopsin in orange. * By photoisomerase activity, we refer to chromophore isomerization from *trans* to *cis* and release to replenish rhodopsin. ^†^ Peropsin was mutated and signaled in the dark; it is likely dark activated. ^‡^ A different cnidopsin from that used in the eyes for phototransduction. Details of evidence and references are in Appendix A. Abbreviations: RGC—retinal ganglion cells, RPE—retinal pigment epithelial, ncPRC—noncephalic photoreceptor cells, PG—pineal gland, PLC—phospholipase C, AC—adenylyl cyclase, GC—guanylyl cyclase, PDE—phosphodiesterase, PIP—phosphatidylinositol 4,5-bisphosphate, TRP—transient receptor potential, CNG—cyclic nucleotide gated, GIRK—G-protein gated inward rectifier potassium, Arr—Arrestin, PKC—protein kinase C, RK—rhodopsin kinase, GRK—G-protein-coupled receptor kinase, UV—ultraviolet, B—blue, LW—long wavelength, S—short, M—medium.

**Figure 2 cells-11-03966-f002:**
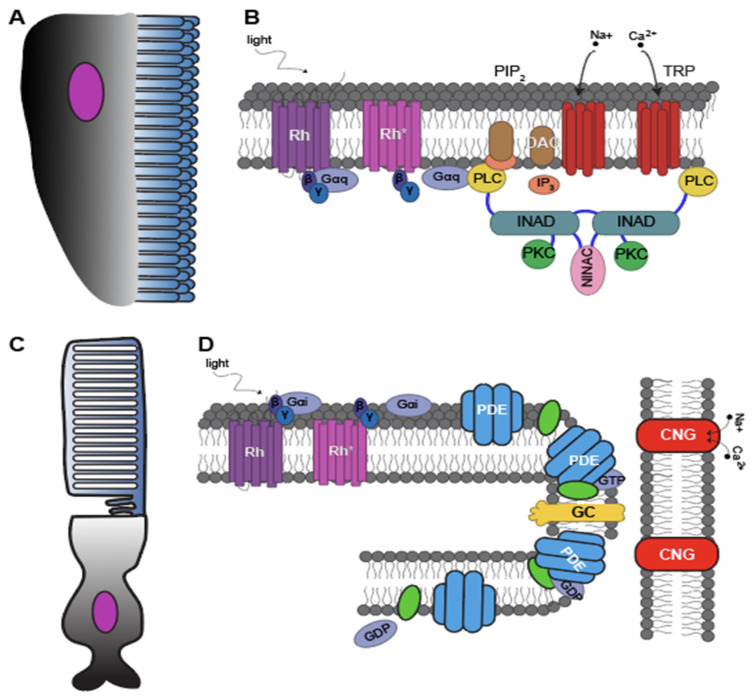
Schematic of rhabdomeric and ciliary receptors and cascades. (**A**) Drawing of a rhabdomeric photoreceptor cell. (**B**) Model of the phototransduction cascade in *Drosophila*. (**C**) Drawing of a ciliary rod photoreceptor. (**D**) Model of the phototransduction cascade in vertebrates. Figure adapted from Fain et al., 2010 [12].

**Figure 3 cells-11-03966-f003:**
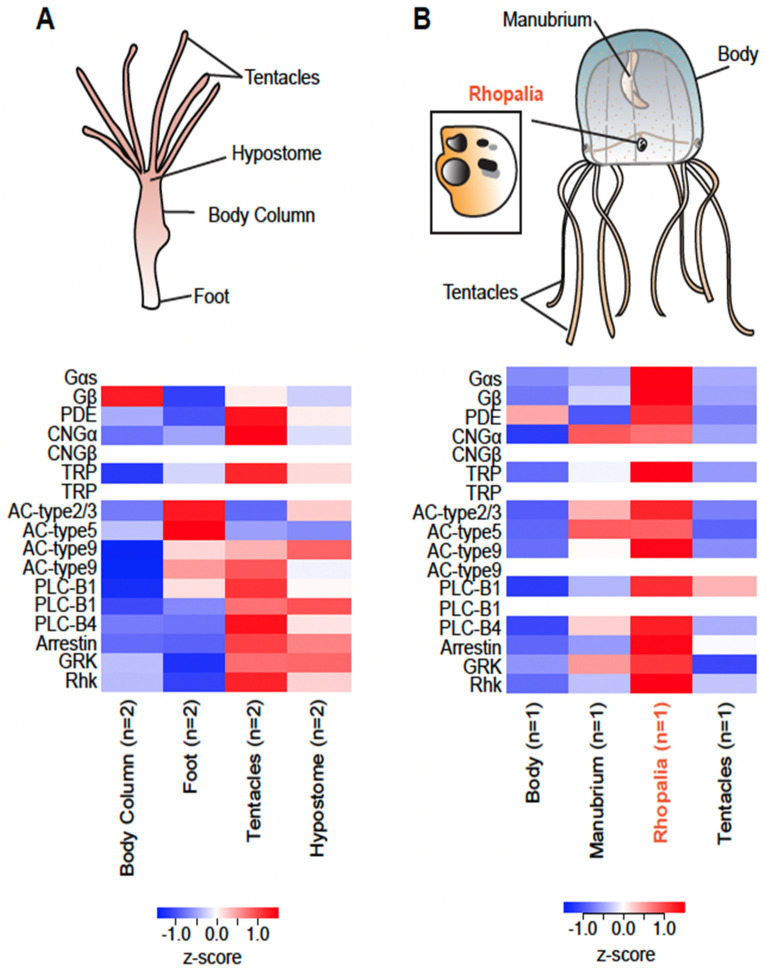
Expression of phototransduction gene homologs in two cnidarian species. (**A**) **Top**: Diagram of a *Hydra vulgaris* polyp. **Bottom**: Heatmap showing relative expression of phototransduction across tissues. Red indicates higher expression in that tissue, and blue represents lower expression. White rows represent missing genes. (**B**) Diagram of *Tripedalia cystophora* and its rhopalium (two lens eye, two pit eyes, and two slit eyes). **Bottom**: Heatmap showing relative expression of phototransduction across tissues.

**Figure 4 cells-11-03966-f004:**
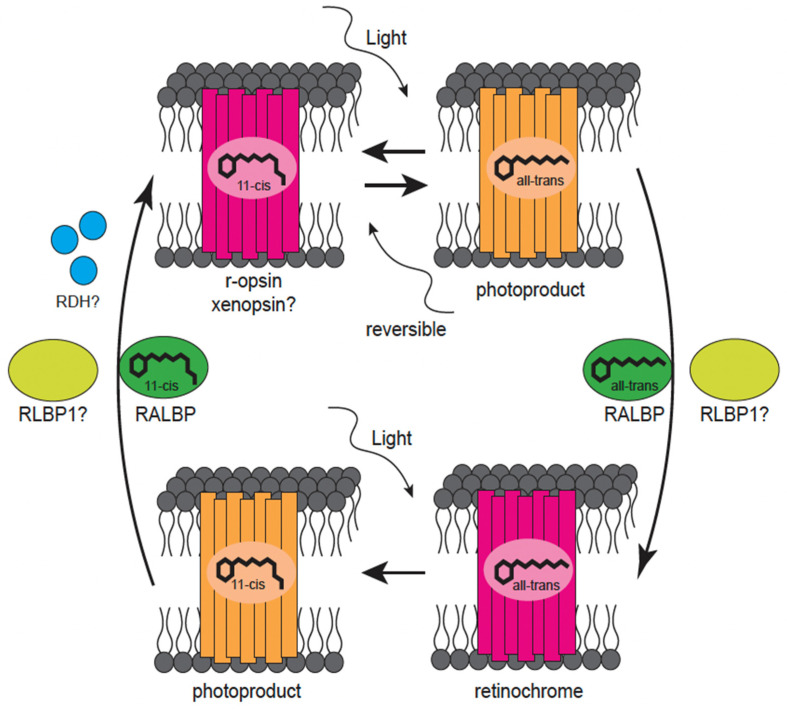
The rhodopsin-retinochrome system. Mostly known from cephalopods, in the rhodopsin-retinochrome system 11-*cis* retinal is bound to an r-opsin and changes its conformation upon light absorption to all-*trans* retinal. Then all-*trans* retinal either remains bound to the bistable r-opsin or leaves the r-opsin and binds to RALBP, which transports it to retinochrome. Bound to retinochrome, retinal changes back to the 11-*cis* conformation upon light detection. Finally, 11-*cis* retinal binds again to RALBP and is transported back to the r-opsin to complete the cycle. However, recent studies indicate more complexity, suggesting the involvement of further proteins, including a potentially monostable xenopsin, the transporter protein RLBP1 and RDH enzymes. However, their actual contributions remain speculative and still need to be determined by functional analyses (adapted from Koyanagi and Terakita 2014 [77]).

**Table 1 cells-11-03966-t001:** Gene copies (potential duplications and loss).

Gene Name	Function	*Hydra*	*Aurelia*	*Tripedalia* *	Reference
G-alpha-i	G-protein cascade activation	1	2	1	Appendix A
G-alpha-o	G-protein cascade activation	1	1	1	Appendix A
G-alpha-q	G-protein cascade activation	1	1	1	Appendix A
GNB1-4-like	member of the G-protein complex	3	1	1	Appendix A
GNB5-like	member of the G-protein complex	1	1	1	Appendix A
CNG-like	calcium/sodium ion channel	0	1	0	Appendix A
CNG-alpha-like	calcium/sodium ion channel	1	1	1	Appendix A
TRP	calcium/sodium ion channel	1	2	1	Appendix A
AC-type2/3	enzyme that transduces G-protein signal	1	1	1	Appendix A
AC-type5	enzyme that transduces G-protein signal	1	1	1	Appendix A
AC-type9	enzyme that transduces G-protein signal	2	1	1	Appendix A
GC 88E	enzyme that transduces G-protein signal	0	3	2	Appendix A
GC-alpha-like	enzyme that transduces G-protein signal	1	1	2	Appendix A
GC-beta	enzyme that transduces G-protein signal	1	1	1	Appendix A
PDE11A	enzyme that transduces G-protein signal	1	1	1	Appendix A
PDE5A	enzyme that transduces G-protein signal	1	1	1	Appendix A
PDE6-like	enzyme that transduces G-protein signal	2	2	1	Appendix A
PLC beta1	enzyme that transduces G-protein signal	3	1	1	Appendix A
PLC beta4	enzyme that transduces G-protein signal	1	1	1	Appendix A
Arrestin	deactivates active rhodopsin	1	1	1	Appendix A
Rhk	deactivates active rhodopsin	1	1	1	Appendix A
SEC14-2	transport protein	1	1	2	Appendix A
SEC14-5	transport protein	1	1	1	Appendix A
Clavesin/RLBP1	transports retinal chromophore	1	1	1	Appendix A
RLBP1-like	transports retinal chromophore	1	1	1	Appendix A
ALDHX	dehydrogenase enzyme	1	1	2	Appendix A
ALDH1/2/X	dehydrogenase enzyme	1	2	2	Appendix A

* This species does not have a genome. Numbers are based on a transcriptome assembly.

## Data Availability

The RNA-seq data presented here are available as follows: *Hydra* is from GEO accession GSE127279 (Murad et al., 2021 [149]) and *Tripedalia* is from NCBI SRR8101518-SRR810526 (Khalturin et al., 2019 [150]).

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
