# Peer review of "Deep Diversity: Extensive Variation in the Components of Complex Visual Systems across Animals"

_cells, 2022, doi:10.3390/cells11243966_

Round 1

Reviewer 1 Report

In evolutionary biology, the idea of "deep homology" was used to describe cases where gene function is conserved across a wide range of species, namely over broad evolutionary timescales.  The authors compare the characteristics of photoreceptor cells, opsins, and phototransduction cascades in different taxa, with a focus on cnidarians.  They propose a "deep diversity" of basic components, showing variation across taxa.

Major point #1 Cnidarians and "deep diversity"

Many reviews, books, and papers discussed the characteristics of photoreceptor cells, opsins, and phototransduction cascades in bilaterians including vertebrates and insects.   However, few discussed Cnidarians on which the authors have worked for many years.  Thus, the authors need to focus on Cindarians and bring up Bilaterians in this context.  However, I felt the authors tried to describe too much about non-cnidarians. I would suggest rearranging the text by focusing on Cindarians and upgrading the authors' idea of "deep diversity"

For example, the authors briefly mention Cnidarians in line 67. Then, in line 90, the authors describe "Photoreceptor cells" and explain photoreceptors in general.  In line 154, the authors finally refer to Cnidarians. However, I would suggest elaborating this paragraph and highlighting Cnidarians.  Birch et al. (Birch et al. In Press) is not in the provided reference list.  I think the readers need to quickly understand the cnidarian visual system without looking at Birch et al., in press or Picciani et al. 2018 etc.  The authors begin to use tentacle bulbs (Sarsia) and rhopalia (Aurelia, Tripedalia) in Figure 3 and section 5.3 (line 443-) without further information.  The authors might want to provide this information (eg, anatomy and hypothetical function).  I also feel a similar issue in the later sections: Opsins and phototransduction cascades (line 169 onward) and Visual Cycle (line 530 onward).  The excellent Cnidarians sections are not well spotlighted.  

Major point #2 Table 1, Figure 3, and Supplementary Figures (Figure S1-S14)

I think that Table 1, Figure 3, and Supplementary Figures (Figure S1-S14) require additional information. How did the authors generate Figure 3 and Figure S1-S14?  I cannot find any references associated with Figure 3. Please explain abbreviations (TPM, Transcripts Per Million, etc).  The image quality of Figure S1-S14 is low.  I would suggest using higher resolution so that readers can read small letters.   

Are these original data? If so, the authors should provide their experimental procedures and write "Results".  More important, I wonder if any articles containing original experimental data can be published as "Review" without citing original papers and evidence.  I can find one clue in its Data availability statement." The RNA-seq data presented here are available as follows: Hydra is 649 from GEO accession GSE127279 (Murad et al. 2021), Tripedalia is from NCBI SRR8101518-SRR810526 650 (Khalturin et al. 2019), and Sarsia and Aurelia will be deposited to NCBI accompanying this paper ".  

Minor points

Line 50-51, line 154 

"eyes may have evolved at lease 40-60 times"

"evolved eyes at least 9 times"

I understand the meaning of these sentences and agree that most readers can understand. However, the expression is too casual to convey the meaning to beginners.

Line 213-230

Here, the authors explain opsins in fish.  

Line 373

The references are italic.

Line 491

eye-bearing tissues (add -)

Author Response

Reviewer 1:

In evolutionary biology, the idea of "deep homology" was used to describe cases where gene function is conserved across a wide range of species, namely over broad evolutionary timescales.  The authors compare the characteristics of photoreceptor cells, opsins, and phototransduction cascades in different taxa, with a focus on cnidarians.  They propose a "deep diversity" of basic components, showing variation across taxa.

Major point #1 Cnidarians and "deep diversity"

Many reviews, books, and papers discussed the characteristics of photoreceptor cells, opsins, and phototransduction cascades in bilaterians including vertebrates and insects.   However, few discussed Cnidarians on which the authors have worked for many years.  Thus, the authors need to focus on Cindarians and bring up Bilaterians in this context.  However, I felt the authors tried to describe too much about non-cnidarians. I would suggest rearranging the text by focusing on Cindarians and upgrading the authors' idea of "deep diversity".

For example, the authors briefly mention Cnidarians in line 67. Then, in line 90, the authors describe "Photoreceptor cells" and explain photoreceptors in general.  In line 154, the authors finally refer to Cnidarians. However, I would suggest elaborating this paragraph and highlighting Cnidarians.  Birch et al. (Birch et al. In Press) is not in the provided reference list.  I think the readers need to quickly understand the cnidarian visual system without looking at Birch et al., in press or Picciani et al. 2018 etc.  The authors begin to use tentacle bulbs (Sarsia) and rhopalia (Aurelia, Tripedalia) in Figure 3 and section 5.3 (line 443-) without further information.  The authors might want to provide this information (eg, anatomy and hypothetical function).  I also feel a similar issue in the later sections: Opsins and phototransduction cascades (line 169 onward) and Visual Cycle (line 530 onward).  The excellent Cnidarians sections are not well spotlighted. 

We have added further information to briefly define the characteristic cnidarian photoreceptive structures. However, since the recent publications of Birch et al as well as Picciani et al give very detailed explanations, we aim to avoid going into too much detail at this point. We use cnidarians here as an example of different eyes and phototransduction cascade to highlight our main themes but refer readers to reviews focused on cnidarian vision. We also feel it is important to review what is known about bilaterian visual systems as part of the comparison with cnidarians.

Major point #2 Table 1, Figure 3, and Supplementary Figures (Figure S1-S14)

I think that Table 1, Figure 3, and Supplementary Figures (Figure S1-S14) require additional information. How did the authors generate Figure 3 and Figure S1-S14?  I cannot find any references associated with Figure 3. Please explain abbreviations (TPM, Transcripts Per Million, etc).  The image quality of Figure S1-S14 is low.  I would suggest using higher resolution so that readers can read small letters.  

Are these original data? If so, the authors should provide their experimental procedures and write "Results".  More important, I wonder if any articles containing original experimental data can be published as "Review" without citing original papers and evidence.  I can find one clue in its Data availability statement." The RNA-seq data presented here are available as follows: Hydra is 649 from GEO accession GSE127279 (Murad et al. 2021), Tripedalia is from NCBI SRR8101518-SRR810526 650 (Khalturin et al. 2019), and Sarsia and Aurelia will be deposited to NCBI accompanying this paper ".

Figures S1-14 are now uploaded as high-resolution PDFs. Methods and Results were added to 5.3. We removed unpublished expression data from Aurelia and Sarsia and focused on results from data that is publicly available and previously published.

Minor points

Line 50-51, line 154

"eyes may have evolved at lease 40-60 times"

"evolved eyes at least 9 times"

I understand the meaning of these sentences and agree that most readers can understand. However, the expression is too casual to convey the meaning to beginners.

We have re-written the respective sentences

Line 213-230

Here, the authors explain opsins in fish. 

We added additional information to this paragraph to make this more clear

Line 373

The references are italic.

We have corrected this mistake

Line 491

eye-bearing tissues (add -)

done

Reviewer 2 Report

This work nicely synthesizes our current understanding of the complexity of photoreceptive systems in the animal kingdom. The text is easy to read, and the primary figures are illustrative and informative.

Comments:

  1. To aid those not as familiar with the various cnidarian eye types being assessed for gene expression, additional references and further description of similarities and differences would be helpful. Reference and incorporation of other physiological and functional studies comparing these eye types is also recommended.

  2. An additional column in Table I that list the role in phototransduction for each listed would be helpful. Sec14, for instance, is only mentioned once in the text, with no description of its function. 

  3. Methods related to the gene expression analysis should be included.

  4. Given the rather methodical listing of information throughout the work, a broader summary and discussion of implications is warranted.  

  5. Minor editing on reference calls in the text are needed.

  6. Low resolution images in the Supplemental File are not readable.

Author Response

Reviewer 2:

This work nicely synthesizes our current understanding of the complexity of photoreceptive systems in the animal kingdom. The text is easy to read, and the primary figures are illustrative and informative.

Comments:

  1. To aid those not as familiar with the various cnidarian eye types being assessed for gene expression, additional references and further description of similarities and differences would be helpful. Reference and incorporation of other physiological and functional studies comparing these eye types is also recommended.

We have added further information to briefly define the characteristic cnidarian photoreceptive structures.

  1. An additional column in Table I that list the role in phototransduction for each listed would be helpful. Sec14, for instance, is only mentioned once in the text, with no description of its function.

Great suggestion, function column was added.

  1. Methods related to the gene expression analysis should be included.

Done. We added section 5.3.1. Methods

  1. Given the rather methodical listing of information throughout the work, a broader summary and discussion of implications is warranted.

We have extended the discussion of our results

  1. Minor editing on reference calls in the text are needed.

We checked our references and corrected the mistakes we found

  1. Low resolution images in the Supplemental File are not readable.

Figures S1-14 are now uploaded as high-resolution PDFs.

Round 2

Reviewer 1 Report

This second version of the manuscript is a great improvement, the authors are to be commended.  

Note: I feel Figures 2 and 3 require better resolution.